# New Insights on the Mobility of Viral and Host Non-Coding RNAs Reveal Extracellular Vesicles as Intriguing Candidate Antiviral Targets

**DOI:** 10.3390/pathogens9110876

**Published:** 2020-10-24

**Authors:** Iwona K. Wower, Terry D. Brandebourg, Jacek Wower

**Affiliations:** Department of Animal Sciences, Auburn University, Auburn, AL 36849, USA; woweriw@auburn.edu (I.K.W.); tdb0006@auburn.edu (T.D.B.)

**Keywords:** lncRNA, miRNA, extracellular vehicles, exosomes, ectasomes, COVID-19, SARS-CoV

## Abstract

Intercellular communication occurring by cell-to-cell contacts and via secreted messengers trafficked through extracellular vehicles is critical for regulating biological functions of multicellular organisms. Recent research has revealed that non-coding RNAs can be found in extracellular vesicles consistent with a functional importance of these molecular vehicles in virus propagation and suggesting that these essential membrane-bound bodies can be highjacked by viruses to promote disease pathogenesis. Newly emerging evidence that coronaviruses generate non-coding RNAs and use extracellular vesicles to facilitate viral pathogenicity may have important implications for the development of effective strategies to combat COVID-19, a disease caused by infection with the novel coronavirus, SARS-CoV-2. This article provides a short overview of our current understanding of the interactions between non-coding RNAs and extracellular vesicles and highlights recent research which supports these interactions as potential therapeutic targets in the development of novel antiviral therapies.

## 1. Introduction

Non-coding RNAs (ncRNAs) constitute ~90% of the human transcriptome. Despite the ubiquitous nature of ncRNA expression, most early research into their functions focused on large ribosomal RNAs (rRNAs). These conserved non-coding components of ribosomes are crucial for the binding of messenger RNAs (mRNAs), recruitment of aminoacylated transfer RNAs (tRNAs), and catalyzation of the peptide bond between two amino acids [1,2]. However, as next-generation sequencing technologies revolutionized the exploration of transcriptomes, a diverse array of novel ncRNAs have been rapidly characterized. It is now apparent that ncRNAs regulate a myriad of important functions in cell differentiation and development and that functional defects in genome-encoded ncRNAs are associated with a wide spectrum of cancers and infectious diseases [3,4].

Non-coding RNAs are arbitrarily categorized into classes comprised of small ncRNAs and long ncRNAs (lncRNAs). Small ncRNAs are a diverse group of molecules composed of less than 200 nucleotides and exemplified by species such as transfer RNAs (tRNAs), tRNA-derived fragments (TFs), small nuclear RNAs (snRNAs), small nucleolar RNAs (snoRNAs), promoter-associated small RNAs (PASRs), PIWI-interacting RNAs (piRNAs), small interfering RNAs (siRNA), and microRNAs (miRNAs). To date, miRNAs and tRNAs are the most extensively studied members of this class of ncRNAs. They are composed of 22 and ~75 ribonucleotides, respectively. In contrast, lncRNAs are very heterogeneous in size with species in this class ranging from several hundred to tens of thousands of ribonucleotides. Currently, GENCODE (v. 35) has annotated 10,671 lncRNA gene loci in humans and these can be transcribed from both the sense and antisense strands to produce tens of thousands of transcripts [5,6]. According to some estimates, more than 60% of transcribed RNAs have antisense complements generated through this mechanism [7,8,9]. 

Importantly, miRNAs and lncRNAs can also serve as potent regulators of the pathogenic mechanisms underlying host–virus interactions. For instance, viruses can successfully compete for the metabolic resources of the host needed to facilitate viral reproduction by exquisitely modifying host cellular metabolism through generating exogenous ncRNAs that cause dysregulated expression of tens to hundreds of host genes involved in metabolic control [10,11]. Additionally, some of the affected genes encode ncRNAs that appear to be important components of the switch to a virus-induced pathogenic transcriptome within the host. Recently Xu and co-workers [12] explored these interactions by RNA-seq analysis of transcriptomes in cells experimentally infected by human foamy virus and identified 4729 lncRNAs that were upregulated and 6588 that were downregulated in response to infection illustrating the significant impact that this virus has upon the host transcriptome. 

A novel mechanism for the transport of viral ncRNAs appears to involve the subversion of membrane-bound extracellular vesicles (EV) that are secreted by eukaryotic cells into the extracellular milieu [13,14]. When first discovered during the 1980s, these EVs, ranging in size from tens to thousands of nanometers, were believed to function primarily in the cellular debris removal. However, it is now known that EVs transport large cargos of proteins, RNAs, lipids, and small molecular regulators and play crucial roles in intercellular communication. More recently, viruses have been shown capable of exploiting EVs to traffic in and out of cells and a rapidly growing literature implicate EVs as important mediators of the mechanisms orchestrating the viral manipulation of the host’s immune system. Intriguingly, newly emerging findings suggest that coronaviruses can generate their own ncRNAs and involve host ncRNAs in virus infection [15,16,17,18,19]. Moreover, cells infected with coronaviruses may produce exosomes that can transfer angiotensin converting enzyme 2 (ACE2), the receptor for the SARS-Cov-2 entry, to other cells and thereby make them susceptible to virus docking [20]. Finally, it has been suggested that exosomes may play an important role in the COVID-19 reinfection [21]. An in vitro study on SARS-CoV-1 cultured in AT2 cells revealed that the virions can be seen within the double membrane vesicles [22]. 

Given the emerging roles of ncRNAs and EV trafficking in viral pathogenesis, especially in the context of the current renewed impetus to search for novel strategies that could prevent global pandemics such as COVID-19, a disease caused by infection with the novel coronavirus, SARS-CoV-2, we review studies that provide a clearer understanding of the interactions between ncRNAs and EVs and highlight the potential that such trafficking may represent a novel target for the development of effective antiviral therapies.

## 2. Emerging Role of miRNAs in Promoting Viral Diseases

The first known miRNA, *lin-4*, was discovered in the nematode *Caenorhabditis elegans* [23]. This 22-nucleotide transcript forms complementary base-pairs with sequences in the 3′ untranslated region (3′UTR) of mRNA encoding the LIN-14 protein. Although the interaction involves only 10 out of 22 nucleotides, the so-called seed sequence, the binding of *lin-4* miRNA to seven sites in the targeted mRNA significantly decreases production of LIN-14 protein and reprograms the timing of larval development. Since, the discovery of *lin-4*, thousands of miRNAs have been characterized in plant, animal, and viral genomes. The miRNA database (miRDB) currently lists 2656 annotated human miRNAs with 1,610,510 gene targets [24]. These miRNAs have been shown to regulate a diverse array of cellular processes during normal eukaryotic cell function by binding to targeted mRNA species and inducing mRNA degradation and/or translational repression [25,26,27,28,29,30,31,32]. Importantly, miRNAs have also been revealed as key mediators of tumorigenesis and cardiovascular, neurological and many infectious diseases [33,34,35,36]. Some viruses (e.g., Kaposi’s sarcoma-associated herpesvirus (KSHV)) have been shown to produce miRNAs that induce glycolysis in the host as a mechanism to acquire necessary energy for successful infection [11,37]. 

A canonical pathway for cellular miRNA biogenesis has evolved as a potent defense against viruses allowing infected organisms to unleash RNA interference (RNAi) mechanisms and modify their non-coding transcriptomes [38]. In this pathway, cellular miRNAs are generated from the primary miRNA (pri-miRNA) transcripts, which are synthesized in the nucleus by RNA polymerase II, with the help of an ATP-dependent RNase named Drosha. The resulting precursor miRNAs (pre-miRNAs) are then exported to the cytoplasm where they are trimmed by another RNase, Dicer, to ~22 nucleotide (nt) double-stranded miRNA duplexes composed of guide and passenger strands. Guide strands associate with multi-subunit protein assemblies known as RNA-induced silencing complexes (RISCs). The guide RNAs recruit the RISC complex to mRNAs with the complementary sequences. Finally, an RNase III component of RISC known as Argonaute (AGO) cleaves the mRNAs targeted by guide RNAs [39].

It is now clear that this canonical pathway can also be subverted by viruses as the host cellular machinery is utilized to produce exogenous, viral miRNAs that appear to be an important component of the mechanism of infection. Many studies demonstrate that viruses use miRNAs to modify both cellular and viral gene expression [40]. Most of them refer to miRNAs encoded by DNA viruses such as adenoviruses, ascoviruses, polyomaviruses, and herpesviruses which have access to the nuclear processing factors. For example, KHSV encodes 25 miRNAs. In KSHV-associated cancers they can account for as much as 20% of all mature miRNA species within an infected cell. Fourteen KSHV-encoded miRNAs can be readily detected in patient exosomes [41] (see Table 1). Given that RNA viruses reproduce in the cytoplasmic space, only a few miRNAs encoded by these viruses are known [18]. As such, viral miRNAs constitute an efficient tool for modifying the host cell environment to enhance virus propagation given a single miRNA may post-transcriptionally regulate tens to hundreds of host protein-coding transcripts and lncRNAs.

Illustrating the importance of miRNA in the mechanism of infection, viral miRNAs can regulate expression of both viral and host genes to coordinately regulate host-virus transcriptomes in ways that allow the virus to successfully suppress the host’s immune defenses. For example, the Epstein–Barr virus (EBV)-encoded BART6 miRNA targets genes of the retinoic acid-inducible gene I (RIG-I) signaling pathway to inhibit the induction of antiviral immune responses [42]. 

Alternatively, through the acquisition of unique binding sites allowing interaction with a specific host miRNA, some viruses mimic the canonical action of endogenous cellular miRNAs within the host as a novel strategy to defeat the host’s immune response whereby normally miRNA binding to 3′UTR would destabilize transcripts or repress translation. For example, binding of the myeloid-cell derived miR-142-3p to the 3′UTR of the mosquito-borne North American eastern equine encephalitis virus (EEEV) is actually associated with reduced induction of the host innate immune response [43]. Less frequently, as exemplified by the interaction of the liver-specific miR-122A RNA with the hepatitis C virus (HCV), miRNAs bind to the 5′UTRs of viral RNA genomes [44]. Another interesting mode of virus-endogenous cellular miRNA interaction takes place during bovine viral diarrhea virus (BVDV) replication. The sequestration of host-expressed miR-17 RNA by this pestivirus confers a functional de-repression of multiple cellular miR-17 targets and extensively modifies the host transcriptome [45]. Most recently, a computational study predicted that six human miRNAs (miR-21-3p, miR-195-5p, miR-16-5p, miR-3065-5p, miR-424-5p, and miR-421) may be able to bind to genomic RNAs of all seven human coronaviruses [15]. Because miR-21-3p expression is upregulated 8-fold in the lungs of mice infected with SARS-CoV-1, such proposed interactions with the virus appear very likely to affect the course of SARS-CoV-2 infection. At present, it is unclear why coronaviruses do not eliminate the binding sites for the host miRNAs. The authors speculate that the miR-21-3p might slow down coronavirus replication in the early stages of infection and thus delay the activation of host’s immune response [15]. Nonetheless, viral interaction with endogenous host miRNA represents an important aspect of viral pathogenesis that warrants much greater attention and study.

In that regard, according to the miRBase depository, miRNAs have been identified in 237 organisms and 34 viruses with most of the listed miRNAs being expressed by DNA viruses [46]. Whether RNA viruses such as coronaviruses and flaviviruses encode miRNAs is controversial and under cautious investigation [47]. Currently most insight into SARS-CoV-1 and SARS-CoV-2-encoded miRNAs has been acquired primarily by computational approaches [17,48]. For example, Fulzele and co-workers found almost 900 human miRNAs that are predicted to bind to genomic RNAs of both SARS-CoV-1 and SARS-CoV-2 [16]. Saçar Demirci and Adan predicted that 30 SARS-CoV-2 miRNAs could potentially target 1367 human genes and that many known human miRNAs may be able to target genes encoding viral structural and non-structural proteins [17]. Such studies suggest a significant interaction between SARS viruses and genomic RNAs in humans. 

Several rodent studies generally point to this link as well. Next-generation deep sequencing studies discovered the presence of three small ncRNAs in lung cells of SARS-CoV-1-infected mice [18]. These three molecules, composed of 18–22 ribonucleotides, are named svRNA-nsp3.1, svRNA-nsp3.2, and svRNA-N as they are derived from the genomic RNA regions encoding SARS-CoV-1 non-structural protein 3 (nsp3) and nucleocapsid protein (N), respectively. Because SARS-CoV-1 replicates in cytoplasm, the svRNAs are believed to be generated by Drosha-independent process. Alternatively, they might be processed by Drosha re-localized to the cytoplasm [49]. Loss-of-function studies using antisense oligonucleotides suggest that svRNAs do not significantly contribute to SARS-CoV-1 replication. However, administration of anti-svRNA-N locked nucleic acids (LNAs) prior to infection with SARS-CoV-1 reduced pulmonary inflammation and production of pro-inflammatory cytokines in mice [18]. The latter observation is very interesting as it suggests that svRNA-N contributes to lung pathology by regulating mRNAs involved in the inflammatory response via the RNAi mechanism. If independently confirmed, the investigations of the svRNA-N roles may lead to novel therapeutic treatments to ease suffering caused by COVID-19.

## 3. lncRNA: Potent Regulators of the Viral Life Cycle and Host Responses to Infection

Some lncRNAs are transcribed by polymerase III, however, the majority of lncRNAs are transcribed by RNA polymerase II [50]. They are frequently 5′-capped, spliced and have 3′-terminal poly(A) tails. In many ways, lncRNAs structurally relate to mRNA except that they do not encode peptides. In infected cells, lncRNAs are often involved in the transcriptional regulation and remodeling of chromosomes, suppressed repression of target mRNAs, and the generation of miRNAs [51,52,53,54]. Moreover, they interact with host proteins and miRNAs, leading to profound yet unexpected effects on virus replication [55,56]. For example, when infecting cells with vesicular stomatitis virus (VSV), Wang et al. identified a lncRNA that, when depleted, inhibited replication not only of VSV, but also herpes simplex virus 1 and vaccinia virus. This study implies that unrelated viruses might use the same host lncRNA to boost their replication. Moreover, it indicates that there are scores of lncRNAs which might use novel unpredictable approaches to enhance their chances of survival in the hostile environment of the host cells [56].

The use of lncRNAs during viral infections is an energy-efficient strategy that likely evolved as an alternative to protein synthesis given the typically small size of viral genomes. Next generation deep sequencing studies indicate that a focus solely on protein coding genes leads to greatly underestimating the true breadth of the human transcriptome. While protein coding transcripts are annotated for only ~2% of genome, a broad spectrum of lncRNAs that do not code for proteins are transcribed across the remaining ~90% of the genome or “dark matter” [57]. It is estimated that the number of lncRNA transcripts produced by human cells could approximate 200,000 [58]. Importantly, the rich sequence information and structural potential collectively inherent in these abundant host and viral lncRNA populations is sufficient to support all mechanisms necessary to regulate cellular processes and viral life cycles either directly as naked RNAs or indirectly via their protein ligands. The interactions between host and viral lncRNAs generally occur in the cytoplasm and the nucleus of the host. However, recent studies demonstrate that these interactions also take place in the extracellular space as lncRNAs are constantly shuttled between cells further suggesting that the biology of these RNA species is diverse enough to accommodate ample opportunity for adaptation in response to evolutionary pressures [59,60,61,62]. 

It is challenging to organize a lncRNA taxonomy of clearly defined, discrete functional categories given the diversity of lncRNA species and their inherent potential to regulate cellular processes. One useful albeit broad strategy is to classify lncRNAs as either infectious or non-infectious species largely based upon their influence upon the host transcriptome. Cytoplasmic lncRNAs generally regulate cellular homeostasis while in contrast, viral lncRNAs dysregulate cellular metabolism and promote an “infection” transcriptomic signature that alters normal cellular function by directing key pathways toward the production of viral progeny within the host independent of normal cellular responses to homeostatic signals [63,64,65]. Readers wishing to learn more about functions of human lncRNAs that are not directly implicated in virus propagation are advised to study excellent reviews by Guttman and Rinn [66], Engreitz et al. [67], Rinn and Chang [68], and Villegas and Zaphiropoulos [6]. 

Viruses have developed sophisticated mechanisms to evade the multi-pronged antiviral responses mounted by the host cell in response to infection. One of the most potent of these viral strategies utilizes error-prone polymerases to synthesize viral lncRNAs that display only modest sequence conservation in contrast to endogenous cellular lncRNAs which are transcribed primarily by the high-fidelity RNA polymerase II [69]. Thus, these viral lncRNAs elude selective pressures endured by cellular lncRNAs. This error-prone “sloppy” transcription mechanism highlights an advantage inherent in the lncRNA-based strategy compared to protein-based approaches. To improve virus viability, viral lncRNAs may both bind to host mRNAs to facilitate their degradation and to viral transcripts to stabilize them [70]. These actions coordinately shut-off host cell protein synthesis and trigger selective translation of viral mRNAs, a coordinate regulation not easily accomplished by protein-based mechanisms.

A diverse array of other evasive viral mechanisms utilizing lncRNAs have been characterized. For instance, some lncRNAs like subgenomic flaviviral RNAs (sfRNAs) play crucial roles in packaging viral genomes and boosting the release of virions [10]. Experiments with yellow fever virus demonstrated that sfRNAs may also play crucial roles in increasing efficiency of viral replication and cytopathicity in infected cells [71]. In another strategy, lncRNAs are used to maintain viral latency and block apoptosis of infected cells [54,72]. Several studies have likewise demonstrated that lncRNAs help in establishing virus-assisted malignant transformation [73,74]. Moreover, viral lncRNAs can impart resistance to the host antiviral responses by affecting differentiation and functions of T cells [75,76]. Finally, it appears that viruses may also be able to direct expression of endogenous lncRNA transcripts encoded by the host genome. While SARS-CoV-2 itself is unlikely to produce lncRNA, researchers discovered 504 differentially regulated lncRNAs in the transcriptome analysis of SARS-CoV-1-infected mouse lung tissue [19]. As our knowledge of lncRNA functions is still very limited, it is presently not clear which of the 504 lncRNAs may help the virus to usurp the host metabolic resources or which of them are being used to defend the host. 

## 4. Extracellular Vesicles: Their Structure and Functions in Intercellular Communication

Efficient and highly controlled cell-to-cell communication is vital to maintain cellular homeostasis and overall physiological health of multicellular organisms. This communication is achieved through diverse cellular mechanisms. The best characterized of these mechanisms involve conveying information either via direct cell-to-cell contacts or through indirect signaling by the local secretion of molecules. In the early 1980s, another communication mechanism was discovered involving extracellular membrane-bound vesicles (EVs) [77,78]. Based upon their sizes and provenance, EVs can be separated into two distinct groups [79]. Exosomes, the smallest group, are approximately 100 nm in diameter while microvesicles (MV) or ectosomes, the largest group, may reach 1000 nm in diameter [13,80]. Though exosomes have endocytic origin, ectosomes are produced by the outward budding of plasma membranes directly into the extracellular space and thus, in contrast to exosomes, the release of ectosomes into the extracellular space does not entail exocytosis [81]. Apoptotic bodies (ABs) and large oncosomes (LO) that are produced by blebbing from apoptotic and non-apoptotic membranes, respectively, are often considered extosomal [82,83,84].

Importantly, almost all cells can secrete EVs, thus, potential viral interaction with EVs is a largely ubiquitous opportunity present across a diverse array of cell types [85]. Given their size, EVs are capable of packaging a broad array of molecules and molecular complexes. The packaging of cargo is EV class dependent as exosome cargo is preferentially selected using diverse processes that involve highly conserved components such as Rab GTPases and the endosomal sorting complexes required for transport (ESCRTs) proteins [86,87]. Recent evidence suggests that at least some ESCRT components participate in the selection and accumulation of cargo components in the ectosome lumen. Once released, EVs can either bind neighboring cells to modify the local microenvironment or travel passively through body fluids such as blood, lymph, and spinal fluid to reach distant target cells, which take them up by multiple mechanisms [88,89,90]. The structure of exosome and ectosome membranes and composition of cargo which these EVs are transporting reflects the metabolic status of cells from which these EVs originated. Recently, some viruses have been shown capable of exploiting EVs to traffic in and out of cells suggesting that EVs serve as important carriers of viral cargo as well [91]. Therefore, both exosomes and ectosomes represent promising targets for research that aims to improve disease diagnosis and drug delivery as well as potential trafficking mechanisms that can be hijacked by opportunistic viruses.

## 5. Circulating Small and Long ncRNAs

The discovery that viruses can exploit exosomes to their benefit initiated a search to identify viral signatures in EVs [92,93]. The best characterized ncRNA species identified as EVs cargo are miRNAs transported by exosomes. The latest release of miRBase, the largest public repository of miRNAs, lists 48,860 mature miRNAs across 271 organisms [94]. Within the human genome, 2654 mature miRNAs have thus far been identified. Computational studies suggest that such a cohort of miRNAs would be sufficient to regulate a majority of human protein-coding genes given that single miRNAs are often predicted to be able to target as many as 300–600 different mRNA transcripts [95]. Most human miRNAs have been implicated in controlling molecular pathways which regulate cellular differentiation, growth, homeostasis, responses to stress, and the initiation of immune defenses against pathogens. Importantly, the list of human miRNAs that have been found in exosomes is growing steadily. In terms of exosomal signatures, the best characterized to date represent exosome-packaged miRNAs that are implicated in tumorigenesis because they may be useful for early cancer diagnosis. Much less is known about miRNAs implicated in viral infection. 

The miRNAs related to viral infection can be divided into two categories. The largest includes miRNAs that are produced by the host in response to viral infection. The second, much smaller category includes virus-encoded miRNAs (vmiRNAs). Work to better understand these miRNAs is hampered by the current lack of public repositories dedicated for vmiRNAs while, unfortunately, the Vir-Mir and VIRmiRNA databases that were established in 2008 and 2014, respectively, are not currently updated by their founders [96,97]. However, despite these structural limitations, advances in our understanding of vmiRNA are being made. For instance, the miRBase (Release 22.1) lists ~200 vmiRNAs encoded in genomes of 21 DNA viruses [94]. Ten human pathogens—Esptein–Barr virus (EBV), Herpes simplex viruses (HSV-1 and HSV-2), Cytomegalovirus (hCMV), Kaposi’s sarcoma-associated herpes virus (KSHV), Herpes B virus (HBV), Torque teno virus 1 (TTV1), human papillomavirus (HPV), human T-lymphotropic virus (HTLV), and polyomaviruses—have been shown to collectively express 67 vmiRNAs. This group is dominated by herpesviruses which after the initial infection often remain latent within specific host cells and may subsequently reactivate. Some of these viruses cause human malignancies (EBV, KSHV), birth defects (hCMV), and a wide array of generally non-overlapping clinical syndromes (HSV1 and 2, HBV, and TTV1). The interplay between cellular miRNAs and HSV-1 and HSV-2 seems to be crucial for establishing latency in infected cells. Furthermore, a human torque teno virus encodes a microRNA that inhibits interferon signaling suggesting that a diverse array of pathogenic functions for vmiRNA will likely be ascribed to vmiRNAs [98].

Importantly, the list of vmiRNAs found in exosomes is also growing. The first vmiRNAs excreted via exosomes were identified in studies involving EBV-infected cells [99,100]. The EBV exploits an miRNA-mediated mechanism to stimulate malignant growth of neighboring cells and weaken the immune system of the infected host and the observation that EBV vmiRNAs are present in exosomes suggests that exosomal trafficking may be one mechanism by which the virus delivers these critical miRNA to the host to facilitate viral propagation. Presently, we know that KSHV [11, 41), HSV-1 [101,102], HSV-2 [103,104], and hCMV [105,106,107] also produce vmiRNAs that can travel in exosomes. Interestingly, KSHV-encoded miR-K12-11 shares a 100% seed sequence homology with human ortholog miR-155. Both miRNAs are able to attenuate TGF-β signaling and thus facilitate viral infection and tumorigenesis [108]. 

Despite their small size, exosomes can entrap whole non-enveloped viruses like hepatitis E virus (HEV) facilitating the non-lytic exit of these viruses and enabling the virus to evade host’s immune system [109]. Moreover, exosomes can incorporate full-length genomic RNA from hepatitis C virus (HCV) and hepatitis A virus (HAV) [110,111,112]. These observations clearly indicate that exosomal trafficking is a viable mechanism that viruses can subvert further supporting the notion that identifying viral signatures in EVs represents an important research focus.

Recent work indicated that lncRNAs can also be present in EVs [60,113]. Here again, most information concerning the transportation of lncRNAs via EVs has been generated from studies aimed at understanding the roles of lncRNAs in neoplastic transformation with the broader goal of developing novel noninvasive diagnostic approaches [61,114]. However, while understanding of the role that EVs-trafficked lncRNAs play in viral infection is lagging, recent research supports such a role. For instance, the trans-activation response element (TAR), a pre-microRNA, and full-length genomic RNAs have all been identified as cargo present in exosomes isolated from HIV patients [115,116,117,118]. Interestingly, exosomes can cross the blood–brain barrier [119]. This raises the possibility that HIV-associated neurocognitive disorders (HAND) which enigmatically are not prevented by antiretroviral therapy may in fact result from an ability of HIV to infect cells in the absence of the CD4 receptor and the chemokine CCR5 and CXCR4 co-receptors by exploiting exosomes to traffic viral agents [120,121,122]. Yet another example of RNA transport facilitated by exosomes involves a tick-borne Langat virus (LGTV), a flavivirus like tick-borne encephalitis virus (TBEV) [62]. This RNA virus induces the release of exosomes from tick cells, which are then taken up by neuronal cells. Interestingly, the infected neuronal cells are also able to release exosomes containing LGTV RNAs. Collectively, these observations support the hypothesis that EVs are able to transport not only small RNAs but also large RNA molecules. Future research is therefore likely to uncover extracellular viral lncRNAs important for pathogenesis.

## 6. Conclusions and Future Prospects

The functions of viral lncRNA, in contrast to vmiRNA, remain largely unexplored. The discovery that viral RNAs can be transported by extracellular vesicles to distant targets in the host organisms provides a new, potentially game changing dimension to our understanding of how viruses conquer the myriad defenses of infected hosts. Given that these new insights on the mobility of viral and host non-coding RNAs reveal extracellular vesicles as intriguing candidate antiviral targets, better understanding the biology underlying EV-based intercellular transport of viral RNAs must now take on greater importance. Doing so may allow the development of novel, effective EV-based diagnostics and therapeutics which allow greater protection against global pandemics. 

## Figures and Tables

**Table 1 pathogens-09-00876-t001:** Virus-encoded miRNAs detected in extracellular vesicles.

Virus	Exosomal Cargo	References
KSHV	miR-K2, miR-K12-1, miR-K12-2, miR-K12-3, miR-K12-3p, miR-K12-4-5p, miR-K12-5, miR-K12-6-3p, miR-K12-6-5p, miR-K12-7, miR-K12-9, miR-K12-10a, miR-K12-10b, miR-K12-11, miR-K12-12	[41]
EBV	BART1, BART2, BART4, BART4, BART5, BART7, BART9, BART11, BART12, BART13, BART16, BHRF1, BHRF1-2, BHRF1-3, BHRF1-5p	[99,100]
HSV1	miRH3, miRH5, miR-H6, miR-H28, miR-H29	[102,123]
hCMV	miR-UL59	[124]

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
