# Peer review of "New Insights on the Mobility of Viral and Host Non-Coding RNAs Reveal Extracellular Vesicles as Intriguing Candidate Antiviral Targets"

_pathogens, 2020, doi:10.3390/pathogens9110876_

Round 1

Reviewer 1 Report

This review highlights an emerging area importance, viral and human ncRNA trafficking and their contribution to the host immune response. It provides a fairly high-level review of the area and notes papers and reviews for a further dive into the literature.  

A few specific comments:

  • I will note that the during the miRNA biogenesis pathway, the miRNA duplex associates with RISC as a single stranded mature miRNA and the final RISC complex not only cleaves target mRNA but it also inhibits protein synthesis in cases of imperfect pairing.
  • Another example of a viral miRNA mimic of a human miR is KSHV-miR-K12-11. This viral miR is an ortholog to hsa-miR-155 and is involved in B-cell transformation.
  • KSHV also produces viral miRNAs that travel in exosomes. (https://doi.org/10.1371/journal.ppat.1003484 and https://doi.org/10.1371/journal.ppat.1006524.
  • Not only can viruses hijack host pathways such as the miRNA biogenesis pathway to produce it's own viral miRNAs, but in many cases mature vncRNAs are found in virions which can have an immediate effect on host gene expression following infection.  
  • A very small number of typos were noted. 

Overall, the topic is of importance and the manuscript covers a large amount of information at a high level while taking dives into the literature in some areas. A general schematic would perhaps be helpful to further illustrate the main message.  

Reviewer 2 Report

This is a concise review describing the interplay between host and viral non-coding RNAs and extracellular vesicles, an interesting topic, as EVs can play a role in virus disease pathogenesis. It would benefit from some modifications, in particular in the introductory section on lncRNAs.

Line 53: explored these 52 interactions by prospecting transcriptomes in cells experimentally infected by human foamy virus. Prospecting needs to be replaced with an appropriate word, the sentence written like this is not clear.

Line 55: illustrating the significant impact that this viral mechanism has upon the host transcriptome. This sentence is unclear, changes in lncRNAs upon virus infection is not a mechanism, but a response. Please reword the sentence.

Line 66: references 14 and 15. Both references are reviews. Unable to find citation for coronavirus in ref.14, and ref.15 is not accessible. They both should be replaced and original reference(s) for coronavirus and EV should be added.

Line 163: an introduction to viral lncRNAs is necessary before discussing their role in cellular functions and viral replication, similar to what the previous sentences do for host cellular lncRNAs. It should include both RNA and DNA viruses, which are the ones better characterized (see reference 54).

Lines 167 and 176 contain statements without an appropriate reference.

The last paragraph of page 4, which continues on page 5, describing the role of some virus lncRNAs in host-cell interactions, is compressed and not very clear. It would benefit from a better discussion of the mechanism(s) by which the viral lncRNAs referenced in the various papers affect host and virus responses. For example, reference 54 refers to the mechanisms of generating miRNAs from a non-coding long viral RNA in neurons.

Line 200: references 57 is not accessible, however neither ref 57 or 58 seem to mention viral lncRNAs in relationship to T-cell differentiation and function.

Reviewer 3 Report

Manuscript” New Insights on the Mobility of Viral and Host Non- Coding RNAs Reveal Extracellular Vesicles as Intriguing Candidate Antiviral Targets “by Wower et al, review the importance of extracellular vesicles in viral propagation and use of these targets for the antiviral treatments. This review article implies like a textbook chapter, with most information on the basics of miRNA, long coding RNA or EVs, and not as a review article that stress on a specific topic like EVs role in antiviral targets. The authors need to focus on the title of the topic and concentrate on the role of EVs in viral and host non-coding RNAs. This is a generalized review article with information on miRNA their biogenesis which are well reviewed in hundreds of review articles.

  1. Line 90-98 , not necessary, as the mechanism of miRNA biogenesis is a well reported, should be replaced with a proper reference.
  2. 99-103, these lines should be modified as few viruses are known to have miRNA, so the specific viruses should be included, rather than generalizing the statement.
  3. Most of the listed miRNAs being expressed by DNA viruses and not in RNA viruses such as SARS-COV-2, and most of the data reported on SARS-COV-2 is from the data base of different sources, that too is not confirmed for COVID-19. The possibilities that these samples are from patients with other complications which also could influence the miRNA profile.
  4. Summarize the data in a table format on DNA viruses and RNA viruses and their miRNAs involving EVs.
  5. Line 206, section 4, is mostly the basic information on EVs, but lack the role of EVs and virus. line 231-235, given a generalized information, and not mention on the specific of viruses that are involved.

Round 2

Reviewer 2 Report

No further revision required

Reviewer 3 Report

Modifications have improved the manuscript significantly.